# Judge's Verdict: A Comprehensive Analysis of LLM Judge Capability Through Human Agreement

## Abstract

This research introduces the **Judge's Verdict Benchmark**, a novel two-step methodology to evaluate Large Language Models (LLMs) as judges for response accuracy evaluation tasks. We assess how well 54 LLMs can replicate human judgment when scoring responses from RAG (Retrieval-Augmented Generation) or Agentic pipelines against ground truth answers. Our methodology progresses from traditional correlation analysis to comprehensive Cohen's Kappa analysis that measures actual agreement patterns. The two-step approach includes: (1) a correlation test that filters judges with strong alignment ($r \geq 0.80$), followed by (2) a human-likeness test using z-scores to identify two distinct judgment patterns—*human-like* judgment ($|z| < 1$) that mimics natural human variation, and *super-consistent* judgment ($z > 1$) that exceeds typical human-to-human agreement levels. This methodology reveals that 27 out of 54 tested LLMs achieve Tier 1 performance: 23 models exhibit human-like patterns that preserve the nuances of human judgment, while 4 models demonstrate super-consistent behavior—a pattern that could indicate either enhanced reliability or oversimplification of complex judgments. Testing 43 open source models (1B-405B parameters) and 11 closed models (GPT, Gemini, Claude variants), we demonstrate that judge excellence is not solely dependent on model size but on specific training strategies. Our key contributions include: (1) establishing that correlation alone is insufficient for judge evaluation, (2) introducing a "Turing Test for judges" based on agreement patterns, and (3) providing a standardized benchmark for classifying LLM judges into distinct performance tiers for different evaluation needs.

## 1 Introduction

The evaluation of AI-generated content traditionally relies on human judgment, which is expensive and time-consuming. This research investigates whether LLMs can serve as reliable judges by conducting a comprehensive two-step analysis comparing 54 different LLM judges against human annotators across 1,994 samples. Specifically, we evaluate how well LLMs can replicate human judgment when both are given the same task: scoring responses from RAGs by comparing them against ground truth answers. This is fundamentally different from evaluating LLM generation capabilities—we're assessing their ability to judge response accuracy in alignment with human evaluators.

This research introduces two fundamental innovations in evaluating LLM judges. First, we move from correlation to agreement: while previous work primarily relied on correlation metrics (e.g., Pearson's $r$) to evaluate judge quality, we demonstrate that correlation alone is insufficient. Our methodology progresses to Cohen's Kappa, which measures actual agreement rather than just linear relationships. This addresses critical issues like systematic bias—an LLM could have perfect correlation while consistently being too harsh or lenient. Second, we design a novel Turing Test for judges based on Cohen's Kappa agreement patterns. Unlike traditional Turing Tests that focus on conversational indistinguishability, our approach asks: "When mixed with human annotators, can we distinguish the LLM from typical human judges?" This test uses z-score analysis of Cohen's Kappa values to identify models that judge like typical human annotators ($|z| < 1$) versus those with exceptional consistency patterns. These innovations establish a more rigorous framework for

validating LLM judgment excellence, moving beyond superficial correlation to actual functional capability, forming a new benchmark—the **Judge's Verdict Benchmark**—that provides a standardized way to assess whether an LLM achieves Tier 1 performance for either human-like evaluation or maximum-consistency tasks.

## 1.1 ANSWER ACCURACY JUDGE

To assess the alignment of each LLM-as-a-judge, we used the Answer Accuracy metric from the RAGAS[1] library which measures how closely a generated answer matches a reference answer. It employs a Large Language Model (LLM) as an evaluator, or "judge," to assess the factual and semantic concordance between the reference and the generated response. The process involves two independent LLM-as-a-judge prompts, each of which are prompted with the user's question, the system-generated answer, and the reference answer in different orders. Each judge assigns a discrete score of: 0 (No Alignment), 2 (Partial Alignment), or 4 (Exact Alignment) 1. To derive the final score, these discrete ratings are first normalized to a continuous scale in 2. The normalized scores of the two judges are then averaged to produce a final accuracy score of the generated answer in 3. This diverse method improves the reliability of the evaluation by mitigating the positional bias and increases the robustness of a single LLM-as-a-judge prompt. More formally, let $S_i \in \{0, 2, 4\}$ denote the discrete score of the LLM judge $i$ comparing the reference and the generated response, where:

$$\text{Judge } i \text{ discrete score } S_i = \begin{cases} 0 & \text{if No alignment} \\ 2 & \text{if Partial alignment} \\ 4 & \text{if Exact alignment} \end{cases} \tag{1}$$

$$\text{Normalization function: } \phi(S_i) = \frac{S_i}{4} \tag{2}$$

$$\text{LLM Judge Answer Accuracy score } = \frac{1}{2}\sum_{i=1}^{2} \phi(S_i) \tag{3}$$

## 1.2 DATASET OVERVIEW

Our dataset combines 6 diverse benchmarks (4 text-based and 2 PDF-based) with 1,994 total samples and 3 expert human annotators per sample (5,982 total annotations). **Expert annotators** were selected through internal data factory from a curated pool of North America-based professionals with bachelor's degrees or higher, relevant domain expertise, native-level English proficiency, and prior RAG evaluation experience. The **data annotation methodology** consisted of three phases: (1) comprehensive onboarding with golden standard examples, (2) a 350-task pilot phase with 100% quality control coverage requiring 80% accuracy threshold for production advancement, and (3) production annotation employing three-annotator consensus with continuous quality control and weekly evaluation sessions. Both expert annotators and LLM judges performed the identical task—comparing responses from RAGs against ground truth answers, assigning scores on a binary scale with partial credit (0, 0.5, 1.0) where 1 indicates the answer fully addresses the question, 0.5 indicates partial coverage, and 0 indicates failure or factual conflicts. **Inter-annotator agreement** was assessed using Fleiss' $\kappa$ with quadratic weighting ($\kappa = 0.79$) and validated using Krippendorff's $\alpha$ with ordinal distance ($\alpha = 0.79$), both indicating substantial agreement and high data quality across all 1,994 tasks.

The data samples are drawn from: **SQuAD v2.0** (346 samples, reading comprehension), **HotPotQA** (342 samples, multi-hop reasoning), **Coral** (318 samples, conversational QA), **TechQA** (295 samples, technical Q&A), **DC767** (347 samples from 767 PDFs with questions across text, tables, charts, and infographics), and **Enterprise-Knowledge RAG (EKRAG)** (346 samples from 5,000 corporate documents including web pages, earnings transcripts, and SEC filings). Answers were generated using Llama-3.1-70B, Llama-3.1-8B, Mixtral-8x22B-Instruct, and Llama-3.1-Nemotron-70B-Instruct

---

[1]https://docs.ragas.io/en/stable/references/metrics/#ragas.metrics.AnswerAccuracy

with three prompts (no context, with retrieved context, and with reference context; detailed in Appendix C) derived from RAGBench Friel et al. (2024), using nvidia/llama-3.2-nv-embedqa-1b-v2 for embedding, nvidia/llama-3.2-nv-rerankqa-1b-v2 for reranking, retrieving top 2 context pieces, chunking by 1024 tokens with 154 tokens overlap, and NV-Ingest[2] for extraction.

## 2 RELATED WORK

Early studies established the *LLM-as-a-judge* paradigm for open-ended NLP evaluation, typically using pairwise preference or direct-scoring prompts and validating by correlation with human judgments. *MT-Bench* and *Chatbot Arena* formalized pairwise judging and surfaced systematic issues such as position and verbosity biases, while showing that strong LLM judges can approximate human preferences on general chat tasks (Zheng et al., 2023). Prompted, rubric-guided judging (e.g., *G-Eval*) improved human alignment in summarization and dialogue via chain-of-thought and form-based prompts, but still reported variability across tasks and metrics (Liu et al., 2023). *Prometheus* (Kim et al., 2024) advanced the field by training a specialized 13B evaluator LLM on synthetic feedback data, achieving Pearson correlation of 0.897 with human evaluators when provided with customized score rubrics—demonstrating that purpose-built judge models can rival GPT-4's evaluation capabilities. *JUDGE-BENCH* ("LLMs instead of Human Judges?") released a 20-dataset suite and found large variance across models, properties, and data types, recommending careful validation against human annotators for each use case (Bavaresco et al., 2024). Complementing preference-style validation, *JudgeBench* constructed correctness-grounded, challenging pairs in knowledge, reasoning, math, and coding, revealing that many strong judges perform only slightly above chance on hard discriminations—highlighting the limits of preference correlation alone (Lin et al., 2024). Beyond text-only settings, *MLLM-as-a-Judge* showed that multi-modal judges align with humans in pairwise comparisons but diverge substantially in absolute scoring and batch ranking, with observable biases and inconsistencies even in frontier models (Chen et al., 2024). Recent work has extended LLM-as-a-judge to domain-specific applications including software engineering (Deshpande et al., 2025; Ashouri et al., 2025), debate evaluation (Wang et al., 2024), content moderation (Rosenfeld et al., 2025), and bias auditing (Ye et al., 2024), revealing domain-specific challenges and biases that inform general judge design.

Our work advances this field by focusing specifically on *response accuracy evaluation* from RAG and Agentic pipelines, moving beyond correlation-based validation to introduce a comprehensive agreement-based framework. We (i) quantify *actual agreement* with humans via Cohen's $\kappa$ rather than just correlation, (ii) introduce a *dynamic, group-based human-likeness test* (a "Turing Test for judges") that identifies human-like versus super-consistent evaluators using $\kappa$-based $z$-scores, and (iii) operationalize a *two-tier excellence criterion* that separates human-like variation from super-consistent reliability. This approach provides a stricter, more nuanced notion of human-level judging than preference-correlation alone, specifically tailored to the practical QA-with-ground-truth setting that underpins many AI applications. By mixing LLM judges with human raters and computing pairwise agreements across 1,994 items from six diverse datasets, we offer a standardized, agreement-centric benchmark that complements existing correlation-based and domain-specific judge evaluations.

## 3 METHODOLOGY

Our analysis measures how well LLM judges align with human annotators when both perform the identical task: evaluating responses from RAGs by comparing them against ground truth answers. The key insight is that we're not measuring LLM performance on generating answers, but rather their ability to judge response accuracy in the same way humans do. To measure LLM's alignment with humans, we employ a progressive two-step approach: correlation analysis followed by comprehensive Cohen's Kappa analysis with human-likeness assessment.

---

[2]https://github.com/NVIDIA/nv-ingest

## 3.1 Two-Step Evaluation Framework

**Step 1: Correlation Analysis** measures the linear relationship between LLM judge Answer Accuracy scores and human consensus scores (averaged across 3 annotators). Correlation is the most intuitive first step—if an LLM judge can't even correlate with human judgments, it's clearly not suitable. High correlation ($r \geq 0.80$) indicates the LLM understands the general pattern of what makes a good vs. bad response. However, correlation has critical limitations: an LLM could have perfect correlation ($r = 1.0$) while being systematically harsh or lenient, correlation only measures relative relationships (not absolute agreement), and it doesn't account for agreement by random chance.

**Step 2: Cohen's Kappa Analysis with Human-Likeness Assessment** addresses these limitations by measuring actual agreement (not just linear relationship) between LLM and human scores, accounting for chance agreement. We employ two complementary approaches: (1) Static Human Baseline Comparison, where we calculate Cohen's Kappa between the LLM and each human individually then average, comparing against a baseline of $\kappa = 0.801$ from human-to-human agreement; and (2) Dynamic Group Analysis (The Turing Test), where we mix the LLM with 3 humans, calculate pairwise Cohen's Kappa between all raters, and use z-score analysis to assess human-likeness: $z = \frac{\kappa_{\text{LLM}} - \mu_{\text{human}}}{\sigma_{\text{human}}}$. Models with $|z| < 1$ demonstrate human-like judgment patterns.

Mathematically, for the Static Baseline:

$$\bar{\kappa}_{\text{LLM}} = \frac{1}{3} \sum_{i=1}^{3} \kappa(\text{LLM}, \text{Human}_i) \tag{4}$$

For the Dynamic Group Analysis with $n = 4$ raters (3 humans + 1 LLM):

$$\mu_{\text{human}} = \frac{1}{3} \sum_{i<j; i,j \in \text{humans}} \kappa(i,j), \quad \sigma_{\text{human}} = \sqrt{\frac{1}{3} \sum_{i<j; i,j \in \text{humans}} (\kappa(i,j) - \mu_{\text{human}})^2} \tag{5}$$

The necessity of both steps is illustrated by two scenarios. In Scenario 1, an LLM might score 0.3 points systematically lower than humans, achieving excellent correlation ($r = 0.95$) but only moderate Cohen's Kappa (0.45) and fails the human-likeness test ($z = -15.2$)—it understands patterns but can't score correctly. In Scenario 2, an LLM scores like a typical human annotator with natural variation, achieving strong correlation ($r = 0.88$), substantial Cohen's Kappa (0.79), and passes the human-likeness test ($z = -0.8$)—demonstrating truly human-level performance.

## 4 Metrics

Our two-step methodology employs specific metrics at each stage, with carefully chosen thresholds based on statistical conventions and empirical analysis.

**Step 1: Pearson Correlation Coefficient** measures the linear relationship between LLM-as-a-judge Answer Accuracy score and human consensus scores, calculated as: $r = \frac{\sum[(x_i - \bar{x})(y_i - \bar{y})]}{\sqrt{\sum(x_i - \bar{x})^2 \times \sum(y_i - \bar{y})^2}}$

We set a threshold of $r \geq 0.80$, representing the boundary for "very strong" correlation following widely accepted conventions (Akoglu, 2018), where 0.80-1.00 is very strong, 0.60-0.79 is strong, 0.30-0.59 is moderate, 0.10-0.29 is weak, and 0.00-0.09 indicates no correlation. This ensures judges demonstrate at least very strong understanding of human judgment patterns.

More formally:

$$\text{Correlation Strength: } r \in \begin{cases} [0.80, 1.00] & \text{Very Strong} \\ [0.60, 0.79] & \text{Strong} \\ [0.30, 0.59] & \text{Moderate} \\ [0.10, 0.29] & \text{Weak} \\ [0.00, 0.09] & \text{No Correlation} \end{cases} \tag{6}$$

$$\text{Required Threshold: } r \geq 0.80 \tag{7}$$

**Step 2: Cohen's Kappa and Z-Score.** Cohen's Kappa ($\kappa$) assesses agreement between two raters accounting for chance agreement, calculated as $\kappa = \frac{P_o - P_e}{1 - P_e}$ where $P_o$ is observed agreement and $P_e$ is expected chance agreement. The Z-score measures how many standard deviations an LLM's average $\kappa$ is from the human mean: $z = \frac{\kappa_{\text{LLM}} - \mu_{\text{human}}}{\sigma_{\text{human}}}$. Our static human baseline shows average human-human $\kappa = 0.801$. According to the Landis & Koch scale (Landis & Koch, 1977), values range from poor ($< 0$) through slight (0.00-0.20), fair (0.21-0.40), moderate (0.41-0.60), substantial (0.61-0.80), to almost perfect (0.81-1.00).

Formally, the agreement calculations are:

$$P_o = \frac{\text{Number of agreements}}{\text{Total number of cases}} \tag{8}$$

$$P_e = \sum_k p_{1k} \cdot p_{2k} \quad \text{(probability of chance agreement for category } k) \tag{9}$$

$$\kappa = \frac{P_o - P_e}{1 - P_e} \in \begin{cases} [0.81, 1.00] & \text{Almost Perfect} \\ [0.61, 0.80] & \text{Substantial} \\ [0.41, 0.60] & \text{Moderate} \\ [0.21, 0.40] & \text{Fair} \\ [0.00, 0.20] & \text{Slight} \\ < 0 & \text{Poor} \end{cases} \tag{10}$$

For identifying human-like judges, we require $|z| < 1$ (within one standard deviation), which captures models that judge like typical human annotators. This is more stringent than traditional statistical significance ($|z| < 1.96$) because our goal is finding models with human-like judgment patterns, not merely avoiding statistical outliers. Notably, all LLMs passing our $|z| < 1$ threshold achieve $\kappa$ values from 0.781 to 0.816, placing them at the boundary between "substantial" and "almost perfect" agreement.

The human-likeness criterion:

$$\text{Human-like if: } |z| = \left| \frac{\kappa_{\text{LLM}} - 0.801}{\sigma_{\text{human}}} \right| < 1 \tag{11}$$

### 4.1 THRESHOLD SENSITIVITY ANALYSIS

To validate our choice of $|z| < 1$ for identifying human-like judges, we analyzed how model classifications change across different z-score thresholds while holding $r \geq 0.80$ constant:

Table 1: Model Classifications Across Z-Score Thresholds

| Z-Score Threshold | Tier 1 Total | Human-like | Super-consistent |
|---|---|---|---|
| $|z| < 0.5$ | 18 | 12 | 6 |
| $|z| < 1.0$ | **27** | **23** | **4** |
| $|z| < 1.5$ | 29 | 29 | 0 |
| $|z| < 1.96$ | 33 | 33 | 0 |

Key observations: (1) At $|z| < 0.5$, we capture only the most human-like judges but exclude many capable models. (2) Our chosen threshold $|z| < 1$ identifies 23 human-like judges and also reveals 4 super-consistent models ($z > 1$). (3) Relaxing to $|z| < 1.5$ or $|z| < 1.96$ adds more models but loses the distinction of super-consistent judges—they get reclassified as human-like. This confirms that $|z| < 1$ meaningfully identifies models that judge like typical humans while preserving the ability to detect exceptional consistency patterns.

## 5 RESULTS

Our two-step evaluation framework progressively filters LLM judges to identify truly human-level performers. We evaluated 54 LLM judges, with 36 passing the correlation threshold ($r \geq 0.80$) and 27 ultimately achieving Tier 1 excellence through our Cohen's Kappa analysis.

Table 2: Top 36 Very Strong Judges Ranked by Correlation

| Rank | Judge | Correlation | Category |
|---|---|---|---|
| 1 | meta-llama/Meta-Llama-3-70B-Instruct | 0.880 | Very strong |
| 2 | mistralai/mixtral-8x22b-instruct-v0.1 | 0.879 | Very strong |
| 3 | google/gemma-3-27b-it | 0.879 | Very strong |
| 4 | gpt-4.5 | 0.874 | Very strong |
| 5 | jondurbin/bagel-34b-v0.2 | 0.872 | Very strong |
| 6 | mistralai/Mistral-Large-Instruct-2407 | 0.870 | Very strong |
| 7 | meta/llama-3.1-70b-instruct | 0.868 | Very strong |
| 8 | meta/llama-3.1-405b-instruct | 0.862 | Very strong |
| 9 | gpt-4.1 | 0.862 | Very strong |
| 10 | meta-llama/Llama-3.3-70B-Instruct | 0.860 | Very strong |
| 11 | Qwen/Qwen2.5-72B-Instruct | 0.858 | Very strong |
| 12 | gemini/gemini-2.5-flash-lite | 0.857 | Very strong |
| 13 | nvidia/llama-3.3-nemotron-super-49b-v1 | 0.852 | Very strong |
| 14 | claude-sonnet-4 | 0.847 | Very strong |
| 15 | Qwen/Qwen3-30B-A3B-Instruct-2507 | 0.846 | Very strong |
| 16 | gemini/gemini-2.0-flash | 0.843 | Very strong |
| 17 | nv-mistralai/mistral-nemo-12b-instruct | 0.842 | Very strong |
| 18 | microsoft/Phi-3.5-MoE-instruct | 0.840 | Very strong |
| 19 | nvidia/llama-3.1-nemotron-70b-instruct | 0.838 | Very strong |
| 20 | openai/gpt-oss-20b | 0.837 | Very strong |
| 21 | meta/llama-4-scout-17b-16e-instruct | 0.834 | Very strong |
| 22 | nvidia/llama-3.1-nemotron-ultra-253b-v1 | 0.833 | Very strong |
| 23 | openai/gpt-oss-120b | 0.833 | Very strong |
| 24 | MaziyarPanahi/calme-3.2-instruct-78b | 0.833 | Very strong |
| 25 | Qwen/Qwen2.5-32B-Instruct | 0.831 | Very strong |
| 26 | jondurbin/bagel-dpo-8x7b-v0.2 | 0.830 | Very strong |
| 27 | nvidia/llama-3.3-nemotron-super-49b-v1.5 | 0.826 | Very strong |
| 28 | mistralai/Mixtral-8x7B-Instruct-v0.1 | 0.823 | Very strong |
| 29 | gpt-4o | 0.818 | Very strong |
| 30 | Qwen/Qwen3-4B-Instruct-2507 | 0.818 | Very strong |
| 31 | gemini/gemini-2.0-flash-lite | 0.813 | Very strong |
| 32 | gpt-4 | 0.811 | Very strong |
| 33 | gpt-5-chat | 0.809 | Very strong |
| 34 | gpt-4o-mini | 0.804 | Very strong |
| 35 | meta/llama-4-maverick-17b-128e-instruct | 0.802 | Very strong |
| 36 | meta/llama-3.1-8b-instruct | 0.800 | Very strong |

While 36 judges achieved very strong correlation ($r \geq 0.80$), high correlation alone doesn't guarantee human-like judgment quality. Our Cohen's Kappa analysis with human-likeness assessment reveals the truly human-level performers. Based on 1,994 items with 3 annotations each, the average human-human Cohen's Kappa is 0.801, establishing our benchmark for "human-level" performance. The dynamic analysis reveals how each LLM's Cohen's Kappa compares against the dynamic human average that varies based on which LLM is in the group.

Table 3: Top 27 Tier 1 Judges Ranked by Consistency (Z-Score)

| Rank | LLM | Cohen's $\kappa$ | Z-Score | Category |
|---|---|---|---|---|
| 1 | mistralai/mixtral-8x22b-instruct-v0.1 | 0.813 | 1.45 | Super-consistent |
| 2 | meta-llama/Meta-Llama-3-70B-Instruct | 0.811 | 1.43 | Super-consistent |
| 3 | google/gemma-3-27b-it | 0.812 | 1.34 | Super-consistent |
| 4 | jondurbin/bagel-34b-v0.2 | 0.804 | 1.01 | Super-consistent |
| 5 | gpt-4.5 | 0.806 | 0.90 | Human-like |
| 6 | meta/llama-3.1-70b-instruct | 0.798 | 0.61 | Human-like |
| 7 | gpt-4.1 | 0.792 | 0.41 | Human-like |
| 8 | meta/llama-3.1-405b-instruct | 0.790 | 0.31 | Human-like |
| 9 | mistralai/Mistral-Large-Instruct-2407 | 0.789 | 0.26 | Human-like |
| 10 | meta-llama/Llama-3.3-70B-Instruct | 0.786 | 0.18 | Human-like |
| 11 | Qwen/Qwen2.5-72B-Instruct | 0.785 | 0.14 | Human-like |
| 12 | Qwen/Qwen3-30B-A3B-Instruct-2507 | 0.780 | -0.04 | Human-like |
| 13 | gemini/gemini-2.5-flash-lite | 0.777 | -0.17 | Human-like |
| 14 | nvidia/llama-3.3-nemotron-super-49b-v1 | 0.775 | -0.20 | Human-like |
| 15 | microsoft/Phi-3.5-MoE-instruct | 0.775 | -0.31 | Human-like |
| 16 | nv-mistralai/mistral-nemo-12b-instruct | 0.774 | -0.39 | Human-like |
| 17 | claude-sonnet-4 | 0.768 | -0.44 | Human-like |
| 18 | gemini/gemini-2.0-flash | 0.769 | -0.44 | Human-like |
| 19 | meta/llama-4-scout-17b-16e-instruct | 0.768 | -0.55 | Human-like |
| 20 | openai/gpt-oss-20b | 0.765 | -0.58 | Human-like |
| 21 | jondurbin/bagel-dpo-8x7b-v0.2 | 0.766 | -0.63 | Human-like |
| 22 | nvidia/llama-3.1-nemotron-ultra-253b-v1 | 0.767 | -0.65 | Human-like |
| 23 | MaziyarPanahi/calme-3.2-instruct-78b | 0.757 | -0.82 | Human-like |
| 24 | openai/gpt-oss-120b | 0.756 | -0.85 | Human-like |
| 25 | nvidia/llama-3.1-nemotron-70b-instruct | 0.756 | -0.87 | Human-like |
| 26 | nvidia/llama-3.3-nemotron-super-49b-v1.5 | 0.762 | -0.94 | Human-like |
| 27 | Qwen/Qwen2.5-32B-Instruct | 0.753 | -0.96 | Human-like |

The 27 Tier 1 judges demonstrate exceptional performance, with multiple models achieving high $\kappa$ scores in the "substantial" to "almost perfect" range per the Landis & Koch scale. The complete

judge performance matrix for all 54 evaluated models, including those that did not achieve Tier 1 status, is provided in Appendix 4. Among these top performers: 4 models are "super-consistent" ($z > 1$) including *mistralai/mixtral-8x22b-instruct-v0.1* ($\kappa = 0.813$), *meta-llama/Meta-Llama-3-70B-Instruct* ($\kappa = 0.811$), *google/gemma-3-27b-it* ($\kappa = 0.812$), and *jondurbin/bagel-34b-v0.2* ($\kappa = 0.804$); and 23 models perform within human range ($|z| < 1$), demonstrating human-like judgment patterns with *Qwen/Qwen3-30B-A3B-Instruct-2507* remarkably close to natural human performance ($|z| = 0.04$).

## 5.1 THE NUANCE-CONSISTENCY TRADE-OFF

Our findings reveal a fundamental tension in LLM judge evaluation: the trade-off between preserving nuanced human judgment and achieving higher inter-rater consistency. Human annotators naturally disagree on edge cases, ambiguous responses, and subjective quality assessments—disagreements that often reflect legitimate differences in interpretation rather than errors. The average human-to-human $\kappa = 0.801$ captures this inherent variability.

The divergence in model behavior raises important questions:

- **Are super-consistent models capturing a "ground truth" that individual humans miss?** One interpretation suggests these models have learned to identify objectively correct answers more reliably than any single human annotator.
- **Or are they oversimplifying complex judgments?** An equally plausible interpretation is that these models achieve high consistency by ignoring the subtleties that cause legitimate human disagreement, potentially missing important edge cases or minority but valid perspectives.

This trade-off has practical implications. In domains where preserving the full spectrum of human judgment is crucial—such as content moderation, creative evaluation, or ethical assessment—human-like models that maintain natural variation may be more appropriate. Conversely, in applications prioritizing reproducibility and consensus—such as standardized testing or compliance checking—the higher consistency of super-consistent models might be valued, with the understanding that some nuance may be lost.

Importantly, our benchmark cannot definitively determine which interpretation is correct. What we can measure is the degree to which models deviate from typical human agreement patterns, leaving users to decide which pattern best serves their specific evaluation needs.

## 5.2 EVALUATION JOURNEY

The evaluation process can be summarized as follows:

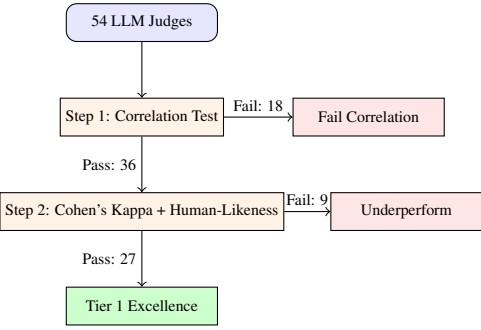

Figure 1: LLM Judge Evaluation Journey

The 27 judges achieving Tier 1 Excellence are divided into two categories. **Tier 1A: Human-Like Judges** ($|z| < 1$) includes 23 models that demonstrate natural human-like variation. Leading models include *Qwen/Qwen3-30B-A3B-Instruct-2507* ($|z| = 0.04$), *Qwen/Qwen2.5-72B-Instruct* ($|z| = 0.14$), *gemini/gemini-2.5-flash-lite* ($|z| = 0.17$), and *meta-llama/Llama-3.3-70B-Instruct*

($|z| = 0.18$). **Tier 1B: Super-Consistent Judges** ($z > 1$) comprises 4 models that exceed human consistency: *mistralai/mixtral-8x22b-instruct-v0.1* ($\kappa = 0.813$, $z = 1.45$), *meta-llama/Meta-Llama-3-70B-Instruct* ($\kappa = 0.811$, $z = 1.43$), *google/gemma-3-27b-it* ($\kappa = 0.812$, $z = 1.34$), and *jondurbin/bagel-34b-v0.2* ($\kappa = 0.804$, $z = 1.01$).

## 6 CONCLUSIONS

Our comprehensive evaluation of 54 LLM judges reveals that high correlation does not guarantee Tier 1 performance. While many judges achieve correlations near 0.9, only 27 out of 54 achieve Tier 1 performance through our two-step methodology: 36 passed the correlation test ($r \geq 0.80$), and 27 ultimately achieved Tier 1 status divided into two distinct patterns—23 Human-Like models ($|z| < 1$) that preserve natural human variation, and 4 Super-Consistent models ($z > 1$) that demonstrate unusually high agreement with human consensus.

Among human-like judges, *Qwen/Qwen3-30B-A3B-Instruct-2507* stands remarkably close to human performance ($|z| = 0.04$), followed by *Qwen/Qwen2.5-72B-Instruct* ($|z| = 0.14$), *gemini/gemini-2.5-flash-lite* ($|z| = 0.17$), *meta-llama/Llama-3.3-70B-Instruct* ($|z| = 0.18$), and *nvidia/llama-3.3-nemotron-super-49b-v1* ($|z| = 0.20$). The super-consistent category is led by *mistralai/mixtral-8x22b-instruct-v0.1* ($\kappa = 0.813$, $z = 1.45$), followed by *meta-llama/Meta-Llama-3-70B-Instruct* ($\kappa = 0.811$, $z = 1.43$), *google/gemma-3-27b-it* ($\kappa = 0.812$, $z = 1.34$), and *jondurbin/bagel-34b-v0.2* ($\kappa = 0.804$, $z = 1.01$).

Notably, model size isn't the determining factor—top human-like models range from 30B to 72B parameters, with closed-source models like Gemini Flash Lite also performing exceptionally well. Architecture, training, and optimization for human alignment matter more than raw parameter count. Our two-step methodology effectively identifies two types of judges from a much larger pool, both achieving Tier 1 performance, but exhibiting fundamentally different agreement patterns that may suit different evaluation philosophies.

This research makes significant methodological contributions. First, we advance beyond correlation by demonstrating empirically why correlation alone is insufficient for judge evaluation. By introducing Cohen's Kappa as a primary metric, we shift the focus from relative patterns to actual agreement. Second, our novel human-likeness assessment creates a Turing Test for judges, establishing the gold standard for human-level performance by identifying models that judge like typical human annotators. Third, this two-step methodology constitutes a new benchmark for evaluating LLM judges—the **Judge's Verdict Benchmark**.

Based on our analysis, the choice between model types depends on your evaluation philosophy and requirements. Human-like models such as *Qwen/Qwen3-30B-A3B-Instruct-2507* ($|z| = 0.04$), *Qwen/Qwen2.5-72B-Instruct* ($|z| = 0.14$), or *gemini/gemini-2.5-flash-lite* ($|z| = 0.17$) preserve the natural variation found in human judgment. Super-consistent models like *mistralai/mixtral-8x22b-instruct-v0.1* ($\kappa = 0.813$) or *google/gemma-3-27b-it* ($\kappa = 0.812$) achieve higher agreement with human consensus than humans typically achieve with each other—a pattern that could indicate either enhanced reliability or oversimplification of nuanced judgments. Models with $|z| > 2$ or $\kappa < 0.61$ show significant deviation from typical human patterns and may not align well with human evaluation standards.

To facilitate research and practical applications in LLM judge evaluation, we provide comprehensive resources as supplementary materials with this submission. The **Judge's Verdict Dataset** contains all evaluation samples from SQuAD v2.0, HotPotQA, TechQA, and EKRAG with 600 human annotations from 3 annotators per sample (200 samples) and complete metadata including source datasets, questions, ground truth answers, and responses generated by RAG/Agentic pipelines. The complete **LLM Judge Evaluation Code** includes scripts for fetching DC767 and Coral source data and evaluating LLM judges on the full dataset. We also provide an **Interactive Leaderboard** with real-time rankings of all evaluated judges, interactive visualization of performance metrics, and separate leaderboards for open and closed source judges. These resources are included as supplementary materials to enable researchers to reproduce our results, evaluate new LLM judges using our methodology, select appropriate judges for specific use cases, and build upon our work to advance the field of LLM evaluation. Upon acceptance, all code and data will be released as open-source resources to benefit the research community.

## 7 FUTURE WORK

While our study provides comprehensive insights into LLM judge performance for response accuracy evaluation, several promising directions remain for future research.

Dataset and annotation expansion represents a critical area for advancement. Our current benchmark includes six diverse datasets, but expanding to additional domains such as medical, legal, and multilingual contexts would enhance the generalizability of our findings. We plan to include specialized domain datasets requiring expert knowledge, add multilingual evaluation datasets to assess judge performance across languages, and incorporate multi-modal datasets as RAG and Agentic systems evolve to handle images and other modalities. Additionally, while our current study involves three annotators per sample, expanding both the number of annotators and annotation volume would enable more robust inter-annotator agreement analysis, allow for annotator expertise stratification, and provide sufficient data for fine-grained error analysis and edge case identification.

Judge model development offers opportunities to create more specialized and efficient evaluators. Our results suggest that smaller models can be competitive judges when properly configured. Future work could explore fine-tuning lightweight models (4B-8B parameters) specifically for response accuracy evaluation, developing domain-specific judge models that excel in particular areas, creating ensemble methods that combine multiple specialized judges, and training judges that can provide detailed feedback beyond binary scores.

A fundamental question raised by our findings is the nature of super-consistent judge behavior. Future research should investigate whether models achieving $z > 1$ are genuinely capturing more reliable judgments or are instead missing important nuances that cause legitimate human disagreement. This could involve: (1) controlled experiments with synthetic data where "ground truth" is unambiguous to test whether super-consistency correlates with accuracy, (2) studies involving domain experts to determine whether super-consistent models align better with expert consensus or lose important minority perspectives, and (3) development of new metrics that can distinguish between beneficial consistency and harmful oversimplification. Understanding this trade-off is crucial for determining when each type of judge model is most appropriate.

## ETHICS STATEMENT

This research involves human annotation work conducted through an internal data factory. All annotators were fairly compensated according to regional standards and provided with comprehensive training and support throughout the annotation process. The annotation tasks involved evaluating publicly available question-answering datasets without exposure to personally identifiable information or sensitive content. Our study design prioritized annotator well-being by implementing reasonable task quotas, regular breaks, and continuous quality feedback rather than punitive measures. We acknowledge potential biases in our evaluation framework, as human judgments may reflect cultural and linguistic perspectives of our North American annotator pool. Future work should expand to more diverse annotator populations to ensure broader applicability of LLM judge evaluations.

## REPRODUCIBILITY STATEMENT

To ensure reproducibility of our results, we provide comprehensive resources as supplementary materials with this submission. Our methodology is fully detailed in Section 3, including precise mathematical formulations for all metrics (Pearson correlation, Cohen's Kappa, and z-score calculations). The experimental setup in Section 4 specifies our dataset composition (1,994 samples across 6 benchmarks), annotation methodology (3 expert annotators per sample), and evaluation framework. All 54 evaluated LLM judges are listed with their exact model identifiers in Tables 1 and 3 and Appendix B. The supplementary materials include: (1) the Judge's Verdict Dataset (2) evaluation code implementing our two-step methodology, and (3) scripts for computing all statistical measures. The threshold sensitivity analysis (Section 3.3) provides justification for our parameter choices. Upon acceptance, all code and data will be released as open-source resources to facilitate replication and extension of our work.

# A TECHNICAL DETAILS

## A.1 STATIC VS DYNAMIC COHEN'S KAPPA

The following diagrams illustrate the key difference between static baseline comparison and dynamic group analysis:

**Static Baseline Comparison:**

- Human baseline is calculated once using only human-to-human comparisons ($\kappa = 0.801$)
- LLM is compared against this fixed baseline
- Simpler but less realistic assessment

**Dynamic Group Analysis (The Turing Test):**

- Human performance metrics change based on which LLM is in the group
- Creates a more realistic "mixed group" scenario
- The LLM must blend in naturally to pass

## A.2 UNDERSTANDING "SUPER-CONSISTENT" MODELS: TWO INTERPRETATIONS

**What "Super-Consistent" Actually Means:**

When a model is labeled as "super-consistent" ($z > 1$), it means the model achieves **higher agreement with human consensus than humans typically achieve with each other**. However, this pattern admits two plausible interpretations that have different implications for practical use.

**Human Annotators Have Natural Disagreement:**

- The average human-to-human Cohen's Kappa is 0.801
- This disagreement often reflects legitimate differences in interpretation, edge cases, and nuanced judgments
- Human variation can capture important subtleties that strict consensus might miss

**Super-Consistent Models Show Different Patterns:**

- mistralai/mixtral-8x22b-instruct-v0.1 ($\kappa = 0.813$, $z = 1.45$)
- meta-llama/Meta-Llama-3-70B-Instruct ($\kappa = 0.811$, $z = 1.43$)
- google/gemma-3-27b-it ($\kappa = 0.812$, $z = 1.34$)
- jondurbin/bagel-34b-v0.2 ($\kappa = 0.804$, $z = 1.01$)

**Two Interpretations of Super-Consistency:**

**Interpretation 1: Enhanced Reliability**

- These models may have learned to identify the "correct" answer more reliably than individual humans
- They could be filtering out human inconsistencies and errors
- This would make them valuable for tasks requiring maximum reproducibility

**Interpretation 2: Oversimplification**

- These models might be missing nuances that cause legitimate human disagreement
- They could be overfitting to the majority view while ignoring valid minority perspectives
- This pattern might indicate less sophisticated judgment rather than superior performance

**The Turing Test Perspective:**

If you mixed 4 annotators (3 humans + 1 LLM) and tried to identify the AI:

- **Super-consistent models**: Would be identifiable due to their unusually high agreement patterns
- **Human-like models** ($|z| < 1$): Would blend naturally with human annotators

**Implications for Different Use Cases:**

The choice between human-like and super-consistent models depends on your specific needs and how you interpret their behavior:

**Human-Like Models** ($|z| < 1$) may be preferred when:

- Preserving nuanced judgments is important
- Natural variation reflects meaningful differences
- The task involves subjective or context-dependent evaluation

**Super-Consistent Models** ($z > 1$) may be preferred when:

- Maximum reproducibility is the primary goal
- The consensus view is deemed most important
- Consistency across evaluations outweighs edge-case handling

**Important Note:** Our benchmark measures agreement patterns but cannot definitively determine whether super-consistency represents enhanced capability or reduced nuance. Users should consider their specific requirements when choosing between these model types.

A.3 MANUSCRIPT PREPARATION

Claude-4.1-Opus was used as a writing assistant to help improve the clarity and readability of this manuscript.

# B    COMPLETE JUDGE PERFORMANCE MATRIX

Table 4: Complete Judge Performance Matrix (All 54 Models)

| Judge | r | $\kappa$ | z | Human-Like? |
|---|---|---|---|---|
| mistralai/mixtral-8x22b-instruct-v0.1 | 0.879 | 0.813 | 1.45 | Super-Consistent |
| meta-llama/Meta-Llama-3-70B-Instruct | 0.88 | 0.811 | 1.43 | Super-Consistent |
| google/gemma-3-27b-it | 0.879 | 0.812 | 1.34 | Super-Consistent |
| jondurbin/bagel-34b-v0.2 | 0.872 | 0.804 | 1.01 | Super-Consistent |
| gpt-4.5 | 0.874 | 0.806 | 0.9 | Yes |
| meta/llama-3.1-70b-instruct | 0.868 | 0.798 | 0.61 | Yes |
| gpt-4.1 | 0.862 | 0.792 | 0.41 | Yes |
| meta/llama-3.1-405b-instruct | 0.862 | 0.79 | 0.31 | Yes |
| mistralai/Mistral-Large-Instruct-2407 | 0.87 | 0.789 | 0.26 | Yes |
| meta-llama/Llama-3.3-70B-Instruct | 0.86 | 0.786 | 0.18 | Yes |
| Qwen/Qwen2.5-72B-Instruct | 0.858 | 0.785 | 0.14 | Yes |
| Qwen/Qwen3-30B-A3B-Instruct-2507 | 0.846 | 0.78 | -0.04 | Yes |
| gemini/gemini-2.5-flash-lite | 0.857 | 0.777 | -0.17 | Yes |
| nvidia/llama-3.3-nemotron-super-49b-v1 | 0.852 | 0.775 | -0.2 | Yes |
| microsoft/Phi-3.5-MoE-instruct | 0.84 | 0.775 | -0.31 | Yes |
| nv-mistralai/mistral-nemo-12b-instruct | 0.842 | 0.774 | -0.39 | Yes |
| claude-sonnet-4 | 0.847 | 0.768 | -0.44 | Yes |
| gemini/gemini-2.0-flash | 0.843 | 0.769 | -0.44 | Yes |
| meta/llama-4-scout-17b-16e-instruct | 0.834 | 0.768 | -0.55 | Yes |
| openai/gpt-oss-20b | 0.837 | 0.765 | -0.58 | Yes |
| jondurbin/bagel-dpo-8x7b-v0.2 | 0.83 | 0.766 | -0.63 | Yes |
| nvidia/llama-3.1-nemotron-ultra-253b-v1 | 0.833 | 0.767 | -0.65 | Yes |
| MaziyarPanahi/calme-3.2-instruct-78b | 0.833 | 0.757 | -0.82 | Yes |
| openai/gpt-oss-120b | 0.833 | 0.756 | -0.85 | Yes |
| nvidia/llama-3.1-nemotron-70b-instruct | 0.838 | 0.756 | -0.87 | Yes |
| nvidia/llama-3.3-nemotron-super-49b-v1.5 | 0.826 | 0.762 | -0.94 | Yes |
| Qwen/Qwen2.5-32B-Instruct | 0.831 | 0.753 | -0.96 | Yes |
| Qwen/Qwen3-4B-Instruct-2507 | 0.818 | 0.749 | -1.26 | No |
| mistralai/Mixtral-8x7B-Instruct-v0.1 | 0.823 | 0.754 | -1.31 | No |
| gpt-4o | 0.818 | 0.728 | -1.55 | No |
| gemini/gemini-2.0-flash-lite | 0.813 | 0.727 | -1.72 | No |
| gpt-4 | 0.811 | 0.723 | -1.73 | No |
| gpt-5-chat | 0.809 | 0.72 | -1.85 | No |
| meta/llama-4-maverick-17b-128e-instruct | 0.802 | 0.712 | -2.18 | No |
| gpt-4o-mini | 0.804 | 0.709 | -2.2 | No |
| Qwen/Qwen3-14B | 0.795 | 0.705 | -2.38 | No |
| gpt-4.1-mini | 0.79 | 0.702 | -2.52 | No |
| mistralai/Devstral-Small-2507 | 0.796 | 0.726 | -2.54 | No |
| meta/llama-3.1-8b-instruct | 0.8 | 0.73 | -2.73 | No |
| CohereLabs/c4ai-command-r7b-12-2024 | 0.793 | 0.724 | -2.88 | No |
| google/gemma-3-12b-it | 0.772 | 0.686 | -2.94 | No |
| microsoft/Phi-mini-MoE-instruct | 0.773 | 0.706 | -3.38 | No |
| microsoft/Phi-4-mini-instruct | 0.778 | 0.719 | -3.56 | No |
| nvidia/llama-3.1-nemotron-nano-8b-v1 | 0.771 | 0.712 | -3.67 | No |
| meta-llama/Meta-Llama-3-8B-Instruct | 0.778 | 0.699 | -4.1 | No |
| google/gemma-2-2b-it | 0.74 | 0.681 | -4.62 | No |
| mistralai/Ministral-8B-Instruct-2410 | 0.762 | 0.663 | -5.01 | No |
| deepseek-ai/DeepSeek-Coder-V2-Lite-Instruct | 0.729 | 0.634 | -6.63 | No |
| meta/llama-3.2-3b-instruct | 0.731 | 0.626 | -7.42 | No |
| Qwen/Qwen2.5-7B-Instruct | 0.67 | 0.532 | -7.74 | No |
| nvidia/nemotron-mini-4b-instruct | 0.671 | 0.615 | -7.76 | No |
| ai21labs/AI21-Jamba-Mini-1.7 | 0.645 | 0.5 | -11.34 | No |
| google/gemma-3-1b-it | 0.614 | 0.56 | -13.49 | No |
| meta/llama-3.2-1b-instruct | 0.02 | 0.005 | -54.74 | No |

**Legend:**

- Super-Consistent: Models that exceed human consistency by more than 1 standard deviation ($z > 1$)
- Yes: Models performing within human range ($|z| <= 1$)
- No: Models that deviate significantly from human patterns ($z < -1$)
- r: Pearson correlation coefficient
- $\kappa$: Cohen's Kappa coefficient
- z: Z-score from dynamic group analysis

## C  PROMPT TEMPLATES FOR RAG RESPONSE GENERATION

We employed three prompts that represent different information access scenarios, derived from the RAGBench framework Friel et al. (2024):

### C.1  NO CONTEXT PROMPT

This baseline prompt template provides only the question without any additional context, testing the model's ability to answer based solely on its pre-trained knowledge:

```
{question}
```

### C.2  CONTEXT-BASED PROMPT (RETRIEVED AND REFERENCE)

Both the Retrieved Contexts and Reference Contexts prompts use the same template structure, differing only in the source of the context documents. This template instructs the model to answer based strictly on the provided context:

```
You are a chatbot providing answers to user queries. You will be given
one or more context documents, and a question. Use the information in
the documents to answer the question.

If the documents do not provide enough information for you to answer
the question, then say ``The documents are missing some of the
information required to answer the question.'' Don't quote any external
knowledge that is not in the documents. Don't try to make up an answer.

Context Documents:
{contexts}

Question: {question}
```

### C.3  KEY DESIGN PRINCIPLES

The context-based prompt template was designed with several important principles:

- **Strict Grounding**: The model is explicitly instructed to use only information from the provided contexts, preventing hallucination or use of parametric knowledge
- **Graceful Failure**: When information is insufficient, the model should acknowledge this rather than generating plausible but unsupported answers
- **Clear Structure**: The prompt clearly separates instructions, context documents, and the question for optimal model comprehension

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
