# OpenReview forum: "Judge's Verdict: A Comprehensive Analysis of LLM Judge Capability Through Human Agreement"
_ICLR.cc/2026/Conference — Submitted to ICLR 2026_

### Official Review · Reviewer_NLJK · 2025-10-26

**Soundness:** 3
**Presentation:** 3
**Contribution:** 3
**Rating:** 6
**Confidence:** 2

**Summary:**

The paper proposes Judge’s Verdict Benchmark, a two-step framework to evaluate LLM-as-a-judge for response-accuracy tasks (e.g., RAG/agent outputs). Step-1 filters models by Pearson correlation r ≥ 0.80 with human consensus; Step-2 computes Cohen’s κ against human raters and performs a “Turing Test for judges” using κ-based z-scores to label models as human-like (|z|<1) or super-consistent (z>1). On 1,994 items with 3 expert annotators each across six datasets, the authors test 54 models, finding 27 Tier-1 judges (23 human-like; 4 super-consistent) and arguing correlation alone is insufficient for judge validation.

**Strengths:**

- Moves beyond correlation to agreement-aware validation (Cohen’s κ) and frames a human-likeness test via κ-based z-scores; this addresses systematic harsh/lenient biases that correlation can miss.
- Broad evaluation: 54 judges, 6 datasets (including PDFs), 1,994 items, 3 expert raters per item; inter-annotator reliability is substantial (Fleiss’ κ≈0.79, Krippendorff’s α≈0.79).
- Clear, actionable tiers (human-like vs super-consistent) that articulate a nuance–consistency trade-off for different application needs.
- Transparent thresholds and sensitivity check for |z|, showing how classifications shift.
- Concrete findings: 36 pass r≥0.80; 27 attain Tier-1; lists of top models by r and κ provided.

**Weaknesses:**

- Grounding of the target task: “Answer Accuracy” is anchored to a RAGAS-style rubric (0/2/4 normalized), but human raters used 0/0.5/1.0—the mapping and potential scale incongruence between LLM judges and humans isn’t fully reconciled (could inflate/deflate κ).
- Threshold choices (r≥0.80; |z|<1) are motivated but still somewhat ad-hoc; no principled decision-theoretic analysis or preregistration, and the baseline human κ=0.801 is treated as near-universal across datasets/labels.
- Potential circularity: humans and models judge the same RAG outputs produced by a small set of models/prompts; this may bias in-distribution to specific answer styles or retrieval artifacts. Dataset generation stack (retrievers/rerankers/chunking) might leak patterns.

**Questions:**

- How exactly are human 0/0.5/1 labels reconciled with the LLM-judge’s 0/2/4→[0,1] scale at scoring time?
- Were judge prompts standardized across models (temperature, system instructions), and are there prompt-leakage risks from using similar instructions in RAGAS and human tasks?
- Do you release code/dataset/leaderboard at review time (with anonymity preserved), and are the thresholds preregistered?

---

### Official Review · Reviewer_Futy · 2025-10-31

**Soundness:** 2
**Presentation:** 1
**Contribution:** 2
**Rating:** 2
**Confidence:** 4

**Summary:**

This paper measures how LLMs can replicate human judgment for RAG-style tasks. To do so, the authors introduce a two-step evaluation protocol, first using Pearson correlation, then using Kohen's kappa. In doing so, they introduce a Turing test for LLM-based judges.

**Strengths:**

1. The evaluation is thorough, spanning 54 LLMs (43 open-source 11 closed-source models).

**Weaknesses:**

1. Many of the citations/references are wrong; they point to correct arXiv identifiers, but the title/authors are hallucinated. For instance, the authors cite "arXiv:2505.16222" as "Testing the limits of llmbased code generation: How well do llms understand code semantics?" by "Erfan Ashouri, Rafael-Michael Karampatsis, and Charles Sutton." The actual article is titled "Don't Judge Code by Its Cover: Exploring Biases in LLM Judges for Code Evaluation" by "Jiwon Moon, Yerin Hwang, Dongryeol Lee, Taegwan Kang, Yongil Kim, and Kyomin Jung." This is not an isolated issue, 8/14 citations have incorrect titles and/or author lists.
2. The writing quality and organization can be greatly improved. For example, Sections 3 and 4 are difficult to distinguish, and Section 5.2 could appear earlier to help the reader understand the evaluation design.
3. The thresholds the authors settle on feel arbitrary (e.g., r>=0.8 or |z|<1). The corresponding interpretations (e.g., "super-consistent" and "human-link") are not well supported.

**Questions:**

See weaknesses above.

---

### Official Review · Reviewer_WVgQ · 2025-11-03

**Soundness:** 2
**Presentation:** 2
**Contribution:** 2
**Rating:** 2
**Confidence:** 3

**Summary:**

The paper attempts to address the problem of unreliable evaluation by LLM judges, where current methods rely on simple correlation with human scores that fail to capture true agreement. A model may correlate well yet consistently rate too high or too low. To address this, the authors propose the Judge’s Verdict Benchmark, a two-step method that first selects models with strong correlation (r ≥ 0.80) and then measures genuine alignment with humans using Cohen’s Kappa and z-scores. This approach distinguishes human-like judges from super-consistent judges, providing a clearer, standardized way to assess whether LLMs truly evaluate like humans.

**Strengths:**

The paper presents a well-defined two-step evaluation method that combines correlation and agreement analysis to measure how closely LLMs align with human judgments. Its structured framework and thorough comparison across 54 models provide clear insights.

**Weaknesses:**

1. No Analysis of Why Existing Methods Fail: The paper criticizes correlation-based approaches but provides no empirical analysis or case studies showing when or why those methods fail. Concrete examples comparing correlation-only evaluations to agreement-based results would make the motivation stronger.

2. Limited Task Scope: Experiments focus mainly on response accuracy in QA/RAG settings. It’s unclear if the framework generalizes to subjective tasks like summarization, dialogue, or creative writing evaluation.

3. Interpretation Ambiguity: The meaning of “super-consistency” is unresolved—does higher agreement mean better reliability or loss of nuance? The paper acknowledges this but doesn’t provide empirical evidence to decide.

4. Presentation: Presentation could be clearer and more concise. The writing is dense, with repeated explanations that make it difficult to follow at times. Figures and tables are informative but not well integrated into the narrative, which reduces readability and flow. A more streamlined structure and clearer visual summaries would make the paper’s main ideas and results easier to grasp.

**Questions:**

1. Could the authors provide more explanation of why previous correlation-based methods fail to meet evaluation requirements? What specific aspects of judgment quality do they overlook?
2. Are there any tasks or datasets where z-scores are notably higher or lower? A breakdown of z-scores by task type would make the analysis more insightful.

---

### Official Review · Reviewer_9vU8 · 2025-11-13

**Soundness:** 2
**Presentation:** 3
**Contribution:** 2
**Rating:** 2
**Confidence:** 3

**Summary:**

The author proposes a novel two-stage benchmark to robustly evaluate the agreement between
human ratings and instruction-following large language models' judgements. The benchmark is
built upon open-access standard LLM benchmarks in an RAG setting. Empirically, 54 LLMs are
evaluated and ranked into different tiers. The main contribution is the inclusion of Cohen’s
Kappa as a metric to capture the nuanced consistency with human ratings.

**Strengths:**

The proposed evaluation method is clear and sound, rooted in classic statistical analysis.

The experimentation comprehensively evaluates the frontier models, including 54 recent
language models.

The paper reveals a phenomenon where human annotators internally and naturally
disagree and discuss the subtle impact on llm evaluation.

**Weaknesses:**

The novelty of the benchmark is limited and insufficiently argued. Although using
Cohen’s Kappa better captures human-llm agreement, the paper fails to address the
importance of adopting the new metric. The example of 2 scenarios presented online
190 is claimed without concrete evidence showing how often it is observed.

The threshold sensitivity analysis fails to provide a rationale for the tier definition. It
appears, the analysis in Sec 4.1 only measures the changes within the selected pool of
language models, without arguing that the chosen models are representative of the
current LLM landscape. There is a likelihood that the chosen thresholds would not
generalize to other language models. Additionally, the definition of thresholds seems to
be based on the number of models falling into each tier, which seems arbitrary and
subjective.

The paper does not include a behavioral analysis to showcase the significance of
“Human-like” and “Super-consistent” models. Models with high correlation scores are
further classified into the two tiers without analysis of how exactly their behavior differs. It
is unclear what being “super-consistent” actually implies about the language model’s
ability and how future model training pipelines could benefit from this metric.

The benchmark setup is not clear and has a limited scope. The questions included are
question answering tests with or without documents with golden answers. A
comprehensive analysis should increase the task diversity and, importantly, include
open-ended creativity-focused tasks. Additionally, there is no analysis of the distribution of annotator scores, explaining how and where they agree or disagree. This provides
evidence for the trade-off discussion in Sec. 5.1.

**Questions:**

How large is the pool, and what is the demographic background of the human
annotators.
How do the low-tier model (with r<.8), human-like model, and super-consistent models
differ in behavior?
Can you show an example where two models have similar correlation but clearly differ in
agreement (κ)?

---

### Meta-Review · Area_Chair_ykvd · 2026-01-11

**Summary:**

The paper proposes a two-step benchmark (“Judge’s Verdict”) to evaluate LLM-as-a-judge for answer-accuracy tasks, arguing that correlation alone is insufficient and adding agreement-aware analysis (e.g., Cohen’s $\kappa$) and a “human-likeness” criterion to categorize judges. Reviewers were largely unconvinced by the novelty and the interpretability of the proposed tiering/thresholding, and noted the evaluation remains limited in scope. A separate major concern is reference integrity: one reviewer reports many citations have correct arxiv IDs but incorrect/hallucinated metadata (titles/authors), which may hint that this paper's citation is largely LLM-generated.. which is serious and unresolved.

**Reviewer Concerns:**

There was no author rebuttal, so key concerns remain outstanding.

**Reviewer Scores:**

Given no rebuttal, I do not expect meaningful score changes under full discussion. All reviewer lean toward rejection.

---

### Decision · Program_Chairs · 2026-01-26

Reject